# Magnetic Tracking of Protein Synthesis in Microfluidic Environments—Challenges and Perspectives

**DOI:** 10.3390/nano9040585

**Published:** 2019-04-09

**Authors:** Melanie Wegener, Inga Ennen, Volker Walhorn, Dario Anselmetti, Andreas Hütten, Karl-Josef Dietz

**Affiliations:** 1Department of Biochemistry and Physiology of Plants, Faculty of Biology, Bielefeld University, D-33615 Bielefeld, Germany; me.wegener@uni-bielefeld.de; 2Center for Spinelectronic Materials and Devices, Faculty of Physics, Bielefeld University, D-33615 Bielefeld, Germany; ennen@physik.uni-bielefeld.de (I.E.); andreas.huetten@uni-bielefeld.de (A.H.); 3Experimental Biophysics and Applied Nanoscience, Institute of Nanoscience BINAS, Faculty of Physics, Bielefeld University, D-33615 Bielefeld, Germany; volker.walhorn@physik.uni-bielefeld.de (V.W.); dario.anselmetti@physik.uni-bielefeld.de (D.A.)

**Keywords:** Arabidopsis thaliana, magnetic field, nanobead, polysome, protein synthesis, ribosome, superparamagnetic particle, translation

## Abstract

A novel technique to study protein synthesis is proposed that uses magnetic nanoparticles in combination with microfluidic devices to achieve new insights into translational regulation. Cellular protein synthesis is an energy-demanding process which is tightly controlled and is dependent on environmental and developmental requirements. Processivity and regulation of protein synthesis as part of the posttranslational nano-machinery has now moved back into the focus of cell biology, since it became apparent that multiple mechanisms are in place for fine-tuning of translation and conditional selection of transcripts. Recent methodological developments, such as ribosome foot printing, propel current research. Here we propose a strategy to open up a new field of labelling, separation, and analysis of specific polysomes using superparamagnetic particles following pharmacological arrest of translation during cell lysis and subsequent analysis. Translation occurs in polysomes, which are assemblies of specific transcripts, associated ribosomes, nascent polypeptides, and other factors. This supramolecular structure allows for unique approaches to selection of polysomes by targeting the specific transcript, ribosomes, or nascent polypeptides. Once labeled with functionalized superparamagnetic particles, such assemblies can be separated in microfluidic devices or magnetic ratchets and quantified. Insights into the dynamics of translation is obtained through quantifying large numbers of ribosomes along different locations of the polysome. Thus, an entire new concept for in vitro, ex vivo, and eventually single cell analysis will be realized and will allow for magnetic tracking of protein synthesis.

Biochemical analyses often suffer from high cost, low throughput, high amounts of required input material, and substandard quality of output. Microfluidic devices present a versatile platform that overcomes the mentioned shortcomings and realizes additional advantages, such as speed of analysis and the minimum requirement of samples. In medical diagnostics, microfluidic systems have proven to be extremely beneficial. Microfluidic devices occasionally are combined with magnetic particles to detect biomarkers in blood or other human liquids [1]. Surface-Enhanced Raman Scattering (SRES) was used to detect poly-γ-D-glutamic-acid (PGA), a marker for Anthrax, with anti-PGA-linked magnetic beads in human serum. The linear detection covered the range from 100 pg/ml to 100 µg/ml, and thus the physiological concentrations [2]. Others used fluorescent antibodies to detect cancer markers after isolating exosomes from human blood serum with antibody-conjugated magnetic particles in microfluidic devices [3].

These examples illustrate the huge potential of microfluidic separation and detection devices. New applications are presently developed and the performance still requires technical improvements in terms of accuracy and detection limit. Therefore, we propose here the detection and quantification of biomarkers in combination with microfluidic devices to study cellular processes ex vivo. Thus, we layout the combination and propose an integration of well-established systems to investigate protein synthesis, in the longer perspective, from single cells, since it is of fundamental interest in the field of modern biology.

## 1. Process and Significance of Protein Synthesis

Protein synthesis is a unique biological process originating from one evolutionarily ancestral mechanism that describes the conversion of the gene-encoded information to the definite protein. The transcription of the gene and the subsequent processing results in the messenger RNA (mRNA), which is decoded into an amino acid sequence by ribosomes in a process called translation. The eukaryotic ribosome is assembled from about 80 polypeptides and ribosomal RNA (rRNA) and displays a molecular mass of about 4.3 MDa [4]. The 3D structure has been elucidated by various methods, e.g., electron cryomicroscopy for the rat 80S ribosome [5]. The ribosome has an oblate shape of about 15 nm (shortest) and 25 nm (longest) diameter [6] and is separated in two subunits. During the initiation phase of translation, the small ribosomal subunit associates with the mRNA and several other proteins (initiation factors), resulting in the initiation complex. The following scanning of the mRNA for the translation initiation codon AUG culminates in the entry of the large ribosomal subunit, and the initiation phase is completed [7,8]. In the second phase of elongation, transfer RNAs (tRNA) specifically charged with amino acids bind to the complementary codon on the mRNA, and the formation of the peptide bond between two neighboring amino acids is catalyzed by the rRNA. In this way the ribosome moves along the mRNA, while the growing peptide chain with the leading N-terminus protrudes to the cytosol through an exit tunnel in the ribosome. After reaching the stop codon, translation is terminated.

A part of many ancillary polypeptides are four major components that are crucial for the process and should be traceable by magnetic means: the specific mRNA associated with the ribosome, the large and small subunit of the ribosomes, and the nascent polypeptide (Figure 1). The complete assembly is called a polysome, since usually many ribosomes simultaneously translate a single mRNA molecule.

Translation is a highly energy demanding process (it consumes up to 30% of the available ATP [10,11]). Therefore, it is not surprising that it is tightly regulated and dependent on environmental and developmental cues [12]. This circumstance is highlighted by a relatively low correlation between transcript and protein amount [13] that goes along with the fact that about 32–48% of the *Arabidopsis thaliana* genome is mainly regulated on a posttranscriptional level [14]. The significance of posttranscriptional regulation is also known in humans, e.g., for many cancer-related oncogenes [15]. This illustrates the urgent need for techniques that address the translational process in specific cells or tissues in dependence on the prevailing environmental, developmental, or pathological conditions.

## 2. Recent Progress in Studying Translation and Polysome Function: Need for Novel Analysis Tools

The toolbox for analyzing protein synthesis in vitro and in cells has been expanded over the last years, even allowing for addressing dynamic processes in the living cell [16]. Profiling of ribosome foot-prints has been developed as a method to determine the dynamics and processivity of ribosomes on RNAs in vivo [17]. Ribosomes are arrested during extraction by addition of inhibitors, such as cycloheximide. RNase treatment of the polysome fraction yields the RNA sites that were covered by the ribosomes in the living cell, which then can be sequenced, aligned, and analyzed. This type of approach also uncovered novel regulatory mechanisms acting on the level of elongation [18].

As ribosomes can also be stalled on RNAs, ribosome association does not necessarily mean that a transcript is translated. Therefore, other methods have been developed to measure on-going translation that are mostly based on the incorporation of traceable amino acid analogues in the nascent polypeptide chain. Some techniques, such as BONCAT (bio-orthogonal noncanonical amino acid tagging) or FUNCAT (fluorescent non-canonical amino acid tagging), use non-canonical amino acid analogues accessible to click chemistry [19,20,21]. Azide labelling with alkylen-conjugated biotin, for example, enables the isolation of de novo synthesized proteins via streptavidin-coated magnetic particles. It is known that pSILAC (pulsed stable isotope-labelling by amino acids) deploys isotopes to differentiate between de novo synthesized proteins that contain heavy amino acids and pre-existing proteins via mass spectrometry analysis [22]. SUnSET (Surface sensing of translation by puromycylation of surface proteins) detects newly translated polypeptides non-radioactively [23] by the premature abortion of translation by incorporating the aminoacyl-tRNA analogue puromycin at the C-terminus of the nascent protein. A part of the immature proteins will reach the cell surface and can be detected and quantified by puromycin antibodies.

At the other end of resolution, optical methods have been developed, which allow for visualization of translation in real time at a single molecular level [24,25,26,27,28,29]. For using TRICK (translating RNA imaging by coat protein knock-off), specific hairpins of the MS2 or PP7 type are introduced frequently in the 3’-untranslated region (3’-UTR) and the coding sequence of the transcript of interest [24]. The corresponding binding proteins, e.g., the MS2 or the PP7 coat protein, are fused to fluorescent proteins and stably expressed in the cell lines. Actively translating ribosomes will kick off the protein and the mRNA will lose its fluorescent color.

Each described method provides important information on ribosome activity and co-translational processes. In this way processivity rates of 3–10 amino acids per second of the nascent protein could be established. Another important finding is the spacing of the ribosomes on the RNA by about 200–300 nucleotides and the translation initiation about every 30 to 40 s [26,27,28,29,30].

However, the described approaches have disadvantages. In the case of TRICK, significant modifications are introduced to the system altering the native state in the cell. The inserted hairpin structures in the UTR or in the coding sequence could influence the initiation phase or the elongation rate of the ribosome. Although TRICK provides single molecule resolution, it is very time consuming to achieve a representative number of experiments. On the other side, ribosome foot printing, quantitative noncanonical amino acid tagging (QuaNCAT), or BONCAT represent averaged results from millions of transcripts and thousands of cells. Therefore, additional, high through-put approaches are required to bridge the averaging methods and single cell sensitivity. We think that magnetic nanoparticles in combination with microfluidic devices present a suitable tool to realize a novel technique for studying translation and to get new insights into translational regulation.

## 3. Magnetic Tracking of Protein Synthesis

Microfluidic devices turned out to be very useful in many contexts, as outlined above. So far, research mostly focused on the detection and quantification of biomarkers. However, the enormous capability of microfluidic devices should also be used to address dynamic processes occurring in single cells as well. Translation is a fundamental process in cells, and translational (de)regulation plays an important role in cancer and other diseases. Thus, we suggest studying protein synthesis in microfluidic environments using tailored superparamagnetic particles. The final goal will be to establish lab-on-a-chip devices enabling magnetic tracking of translation from single cells. The envisioned device should combine lysis of a selected single cell, labelling of selected polysomes, separation, detection, and quantification. The realization of such novel approaches requires four developments: (i) production of superparamagnetic particles of sufficiently small size, (ii) their functionalization for specific binding to the biomolecule of interest, (iii) the separation in microfluidic devices, and (iv) the detection and analysis of the isolated structures.

### 3.1. Functionalization and Specific Labelling

To label polysomal assemblies one can revert to a broad range of commercially available magnetic particles that allow for immobilization of various biological probes. Streptavidin functionalization enables the binding of any biotinylated molecule, such as sequence-specific oligonucleotides. Antibodies can be bound by Protein A-coated particles, while amine-functionalized particles can be used to track proteins in general [31]. Many strategies are established for tailored functionalization and are also realized in commercially available magnetic beads, such as MyOne/Dynabeads of about 1µm diameter (ThermoFisher, Waltham, MA, USA), or other beads of a size of 3.5 µm for magnetic-activated cell sorting (MACS, Miltenyi Biotec, Bergisch Gladbach, Germany). As the translation machinery presents a supramolecular nanostructure composed of different biomolecules, various tagging strategies can be applied to address the different components of the polysome; the transcript, the ribosome, or the nascent protein.

The mRNA component of the polysome can be targeted by complementary DNA-oligonucleotides that hybridize with the target RNA-sequence (Figure 2A). Depending on the used DNA probe, one can track all poly-A-tailed mRNAs with oligo-dT-probes or single transcripts using DNA-probes that address unique RNA-sequences. An own example is shown in Figure 2A. Oligo dT-labelled magnetic particles were used to pull down poly-A-tailed polysomes from crude *Arabidopsis thaliana* extract in the presence of cycloheximide. After RNA extraction from the bead fraction (dT-probe) and cDNA-synthesis, amplification of both *ACT2* and 18S rRNA by reverse transcription polymerase chain reaction (RT PCR) hint to successful affinity isolation of polysomes. The appropriate controls with biotin blocked magnetic particles (biotin) or the no-template control of the PCR reaction showed no or less signal, whereas the positive control (+, RNA extracted from plant extract) shows the expected strong signal.

Ribosomes can be targeted by IgG-functionalized particles. One has to select ribosomal proteins that are localized on the surface, and therefore, are accessible for the appropriate antibody. The first experiments with anti S14 antibody and anti-rabbit immunoglobulin (IgG)-coated gold particles resulted in a successful immunogold labelling of the small ribosomal subunit, and thus polysomes (Figure 2B).

Like ribosome-associated polypeptides, the newly synthesized peptide chain can be captured using specific antibodies, as depicted in Figure 2C. Polyclonal antibodies prepared against denatured proteins should contain IgGs that recognize epitopes of the nascent protein. Alternatively, epitope-specific antibodies should be generated against peptide stretches of the N-terminus. To better match the size of small nascent peptides, one can revert to single-domain antibodies, so called nanobodies, or aptamers. Both molecules show similar affinities to their targets, like conventional antibodies, but benefit from small size, high stability, and relatively cheap production. As every polysome contains all three components, multiple labelling is possible, so that different strategies can be combined optionally, depending on the aim of study. Applying different probes (for example, DNA-oligonucleotides to select a transcript and antibodies to label every ribosome on that transcript) at once or stepwise, one can enhance the specificity for the selection of the desired structure. One would not just be able to isolate a transcript of interest but also to address specific assemblies, such as a selected transcript that carries a specific number of ribosomes, or perform “molecule sorting” depending on the number of associated ribosomes. The additional decoration of nascent proteins with fluorescent nanobodies would allow for identification of active ribosomes that carry nascent peptides using optical sensors.

The suitability of immune-based binding regimes for selection and separation is established in methodological protocols, such as RNA immunoprecipitation (RIP), which allows the isolation of selected mRNP-assemblies using immobilized antibodies directed against RNA-binding proteins [32], or in chromatin immunoprecipitation (ChIP) approaches [33]. Also, specific ribonucleoprotein complexes were extracted using target-specific complementary nucleotide sequences in yeast [34]. The aim will be to combine these techniques with microfluidic platforms in order to down-scale the sample input and study single cells.

### 3.2. Proposing a New Device for Separation and Detection Ensuring Further Analysis

The ultimate goal is to implement the discussed labelling strategies in microfluidic systems and to combine them optionally to achieve automated molecule separation depending on the association with magnetic particles, as sketched in Figure 3. Once a cell is inserted into the microfluidic channel system it will be lysed by chemicals, such as detergents or mechanical forces. From our experience, we do not expect any negative effect of the buffer components on the subsequent labelling procedure. In the beginning, plant protoplasts could be used as a model, as these wall-free cells can be lysed easily by hypoosmotic shock or digitonin [35,36]. The released compounds are then accessible and the molecule assembly of interest can be targeted using specific particle-bound probes. Cycloheximide will be provided during cell lysis to stall ribosomes on the mRNA, and in this way freeze the current translation state of the cell. The use of such translation inhibitors stabilizes the supramolecular assemblies and minimizes alterations of the ribosome-RNA interactions during the following labelling and separation procedure. Labelling efficiency will be enhanced by mixing cell ingredients and functionalized particles in reaction chambers that are integrated in the microfluidic channel.

To be able to analyze the assemblies of interest, it is essential to separate bound and unbound molecules. This seems simple for single labelled molecules that can be collected by magnets and analyzed. Unlike this, the separation of complexes that have been labelled multiple times is not that straight forward and requires a huge effort. To separate polysomal assemblies that differ in ribosome occupancy, we envision integrating magnetic ratchets so as to benefit from all intrinsic physical advantages present. Manipulation, separation, and characterization of biological cell constituents based on magnetic ratchet effects involve the combination of superparamagnetic nanoparticles as markers for biomolecules, controlled transport mechanisms to separate and sort them, and biochemical means to investigate the separated biomolecules after removing them from the microfluidics.

One of the simplest theoretical ratchet concepts is the so-called on–off ratchet. A spatially periodic, asymmetric magnetic potential in one dimension, sketched in Figure 4A, is periodically switched on and off over time. When the potential is turned on, particles are trapped in the vicinity of its minima, whereas they diffuse freely when it is turned off. Due to the asymmetric magnetic potential shape, particles are more likely to cross the barrier towards the next minimum in one direction than in the other during the diffusive phase. This leads to an average net movement in the preferred direction. Magnetic ratchets, thus, make it possible to manipulate colloidal suspensions in micro- or nanofluidic channels and have slowly found their way into biotechnology as a separation device [37,38].

Beads loaded with biomolecules experience higher friction due to their increased hydrodynamic diameter than those without biomolecules, and therefore move more slowly through the ratchet (see Figure 4B)**.** This ratchet concept will allow for sorting polysomes regarding ribosome occupancy. Preliminary tests with standardized molecules will give hints as to the time certain molecules of a certain size will reach a certain position. Outlets in the magnetic ratchet will enable the harvesting of polysomes that can be analyzed by biochemical or molecular biological approaches or physical sensors, afterwards allowing for quantification. A pre-sorting step via magnetic gradients, as indicated in Figure 3, step 4, prior to separation with magnetic ratchets, will reduce the likelihood for deposition of waste and cell debris and guarantee a fluent trouble-free separation process. The exact output would be, for example, how many transcripts of a certain gene carry one ribosome, how many carry 5 ribosomes, how many carry 10, and so on, so that it is possible to generate a distribution for ribosome occupancy for specific transcripts and compare this for different treatments, cell types, or developmental states.

### 3.3. Microfluidic Design and Physical Parameters

With our goal being to employ superparamagnetic nanoparticles that match the size of polysomes, we will experience a paradigm shift in applying microfluidics. Since mixing under laminar flow conditions for large entities ≥ 1 µm can only be realized by diffusion of beads and polysomes for mutual bonding, our microfluidic device in Figure 3 foresees reaction chambers, where the laminar flow lines are narrowed, promoting diffusive mixing of beads and polysomes. With smaller nanoparticles, we expect an enhanced bonding probability with polysomes due to their much larger Brownian motion.

The microfluidic device will be fabricated utilizing 3D digital light processing printing, which makes use of a liquid photopolymer resin, which solidifies under a light source. In this way we will realize a casting mold, which will be molded with polydimethylsiloxane (PDMS) in a second step. The resulting microfluidic channel width will be 120 µm, aiming for a channel height of 20 µm. This relatively small channel height is required to bounce back fast, diffusing small nanoparticles into the influence of laminar flow and within the range of the ratchet into the magnetic gradient fields.

On a Si-wafer chip with a thermalized SiO-surface, the conducting lines of the ratchet are produced by standard optical lithography methods. The microfluidic PDMS-channel will then be attached to the Si-wafer chip, employing an oxygen plasma for activating the PDMS in order to realize strong bonding onto the chip [38]. After integrating the pumps and finalizing the electrical bonds for operating the ratchet, our proposed device is ready.

Concerning flow velocities, we are expecting velocities of about *u* = 250 µm/s for large entities with about 1µm in size within the microfluidic channels. The velocities for smaller nanoparticles are expected to be higher. The velocities within the ratchet are determined by the on–off frequency. Depending on the operation process of the ratchet, polysomes bound to nanoparticles and nanoparticles without any load can be separated bidirectionally, assuming *u* = 0 µm/s. The interacting on–off frequency and flow velocity with *u* > 0 µm/s is one of the challenges to be tackled.

### 3.4. Challenges and Perspectives

Magnetic tracking of protein synthesis is in reach if several of the described parameters are optimized and the described methods successfully combined. One critical feature of the approach is the size of the superparamagnetic particles and their functionalization. The size of the particles should be reduced, and should ultimately match the dimension of ribosomes with about 25 nm. Commercially available magnetic particles range from 2.8 µm to 200 nm, in some cases 100 nm. Diverse synthetic regimes are at hand to produce nanoparticles smaller than 5 nm [39].

Additionally, magnetic beads or nanoparticles in a microfluidic environment require a large susceptibility of the magnetic carriers used, so that they can be aligned by relatively small external fields. A large susceptibility is usually not feasible with commercially available beads, since these are composed of many very small superparamagnetic nanoparticles, usually Fe_2_O_3_ or Fe_3_O_4,_ and thus characterized by the mean magnetic properties of the enclosed nanoparticles. Moreover, ideal nanoparticles should be characterized by a large saturation magnetization, which will lead to a high magnetic moment, enabling their movement in external magnetic potential landscapes. The interplay between the size of superparamagnetic nanoparticles and their magnetization reversal is illustrated for different magnetic phases in Figure 5. Only from a target size of more than 25 nm diameter do the magnetization curves show a high susceptibility. Despite a high saturation magnetization, this is not the case with smaller diameters. Thus, possible magnetic phases should be characterized by a superparamagnetic limit that is in the range of ≥ 25 nm.

By comparing the working range of possible superparamagnetic phases, suitable nanoparticle systems for labelling and separating polysomes are identified in Figure 6. These nanoparticles can be fabricated following either a bottom-up approach by chemical means (nanoparticles within the rectangular area, marked with a solid black line) or a top-down access relying on e-beam lithography (within the rectangular area, marked with a dashed black line). In particular, Heusler alloy-based nanoparticles [40] (marked with pink stars) lead to the desired material property combining high moment and high susceptibility. The pink line indicates the size limit for nanoparticles to result in a large susceptibility.

Moreover, all these potential superparamagnetic nanoparticles match the size of polysomes and will open a new avenue for labelling in a way that ideally each particle only binds to a single polysome. Currently, this seems problematic, as so far hundreds or thousands of probes are conjugated to one bead. Nevertheless, the stoichiometry could be adjusted by adding excess amounts of particles to the sample, or by reducing the number of binding sites on each particle by adding competitive blocking molecules.

To bridge the gap between these nanoparticles of about 25 nm in diameter towards several 100 nm ones, we propose an alternative bottom-up approach, employing a Dual-Beam Focused Ion Beam (FIB). Co_2_(Co)_8_ as a metalorganic precursor can be decomposed within the FIB and the resulting Co-atoms can directly be deposited as nanoparticles within the size range of 25 nm up to about 300 nm. Due to the high C-content of 40%, which is related to this fabrication process, these nanoparticles are characterized by superparamagnetic behavior over the entire size range, with a magnetization of 852 kA/m, which is much higher than that of magnetite.

Ultimately, applying magnetic particles to polysomes in microfluidic environments will allow researchers to gain new insight into translational regulation under different conditions. This method promises new benchmarks in terms of resolution, throughput, and required input material. Separating single transcripts according to their ribosome occupancy enables ribosome counting. The high selectivity of magnetic ratchets enables polysome separation in native conditions with a much higher resolution than sucrose gradient centrifugation, which results in sucrose-contaminated polysomal fractions. Unlike methods such as TRICK, this approach will not provide real time information about translation kinetics, as the assemblies are analyzed after arrest of the translational process with cycloheximide.

The experimental design will have to be optimized, e.g., one can chose between labelling with several superparamagnetic nanoparticles or a single nanoparticle combined with non-magnetic labels. The selection of the labelling strategy will have to be harmonized with the separation and detection method used in the microfluidic design.

Magnetic tracking of protein synthesis will give quantitative information on the translation of specific transcripts in the cells prior to their lysis and is dependent on the environment, disease, stress, or pharmacological treatments. No prior modification of the biosystem, such the insertion of special sequences for structure formation in the UTRs, is required for the proposed in situ tracking. The use of the microfluidic environment is of advantage, as all steps are performed in a single closed system once the cell is inserted. This minimizes possible contamination with disturbing substances, such as RNases, which is a big threat for all work with RNA, including polysomes. Naturally, the most important novelty comes from requirement of low material input.

Major efforts are required to manage the experimental challenges and to realize such a lab-on-a-chip-device. However, this is worthwhile, since the proposed approach opens up a compelling perspective for novel insight into protein synthesis.

## Figures and Tables

**Figure 1 nanomaterials-09-00585-f001:**
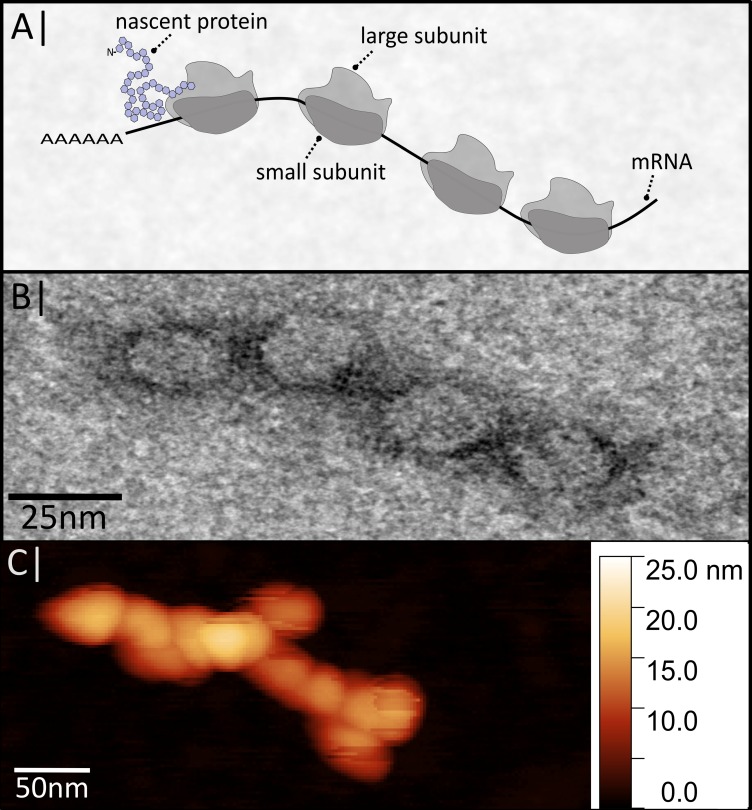
(**A**) Schematic supramolecular structure of a polysome. (**B**) Electron microscopic image of a polysome. (**C**) Polysomes imaged by atomic force microscopy. Shown micrographs are own results. See Appendix A Material and Methods A1 for detailed information [9].

**Figure 2 nanomaterials-09-00585-f002:**
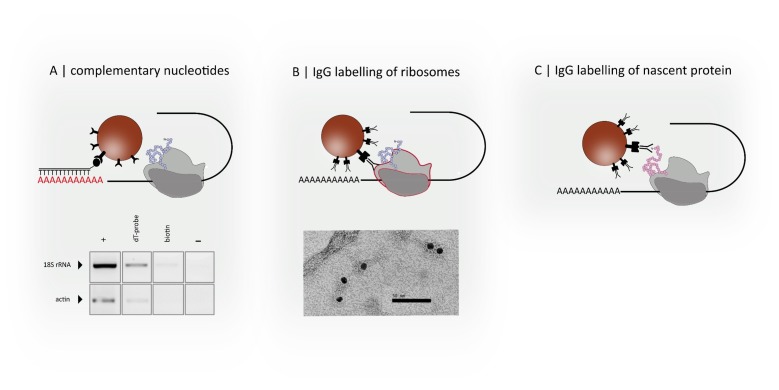
(**A**) Labelling and isolation of specific transcripts by complementary nucleotide sequences. (**B**) Labelling of ribosomes with antibodies (IgG) and immunogold labelling of polysomes. (**C**) Labelling of nascent proteins with antibodies. Shown results are own data. See Appendix A Material and Methods A2 and A3 for detailed information.

**Figure 3 nanomaterials-09-00585-f003:**
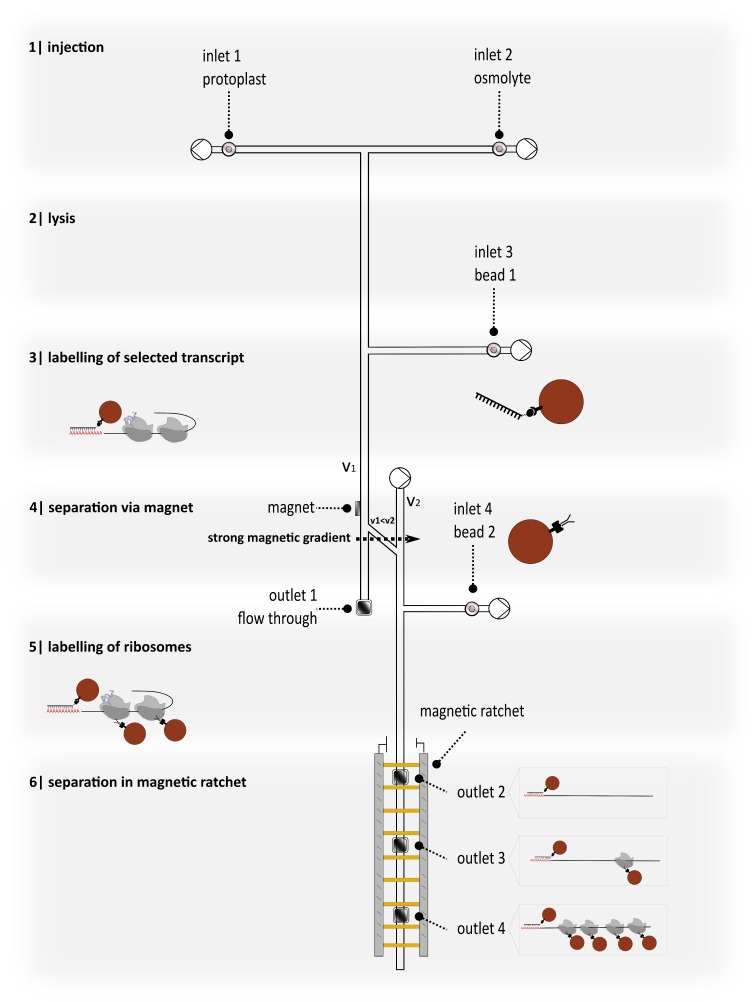
Schematics of microfluidic separation of differentially labelled polysomes, including cell lysis, magnetic labelling, mixing, separation, and detection.

**Figure 4 nanomaterials-09-00585-f004:**
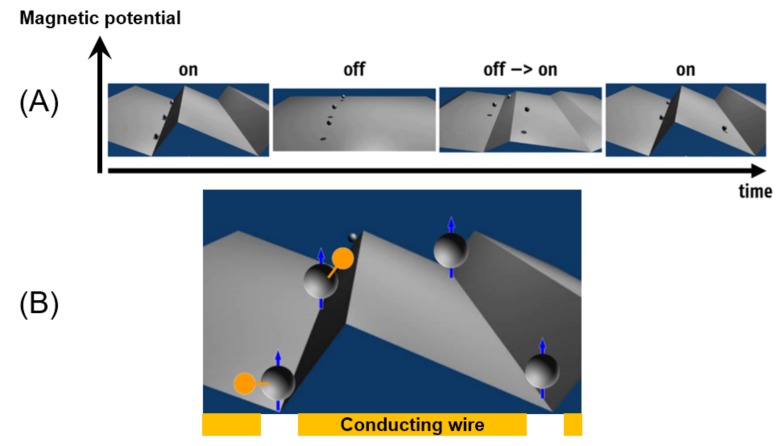
(**A**) Sketch of the ratchet dynamic over time. (**B**) Schematics of a magnetic on–off ratchet in the on-state with magnetic carriers (gray) with and without associated biomolecules (orange).

**Figure 5 nanomaterials-09-00585-f005:**
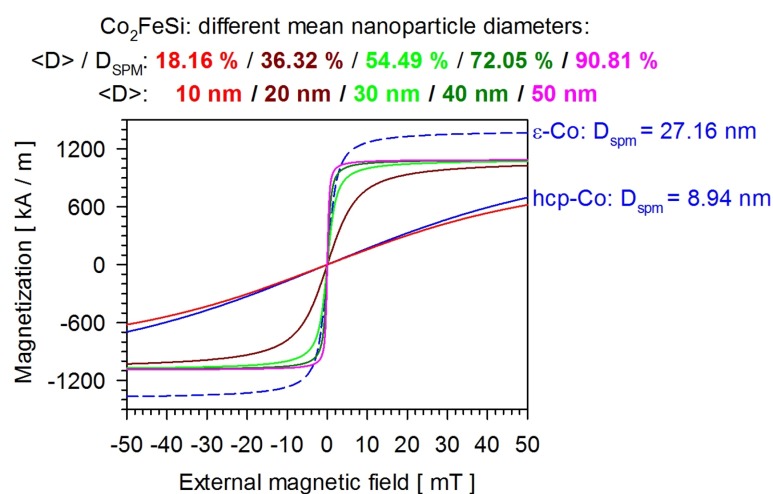
Calculated magnetization reversals of superparamagnetic nanoparticles of different sizes.

**Figure 6 nanomaterials-09-00585-f006:**
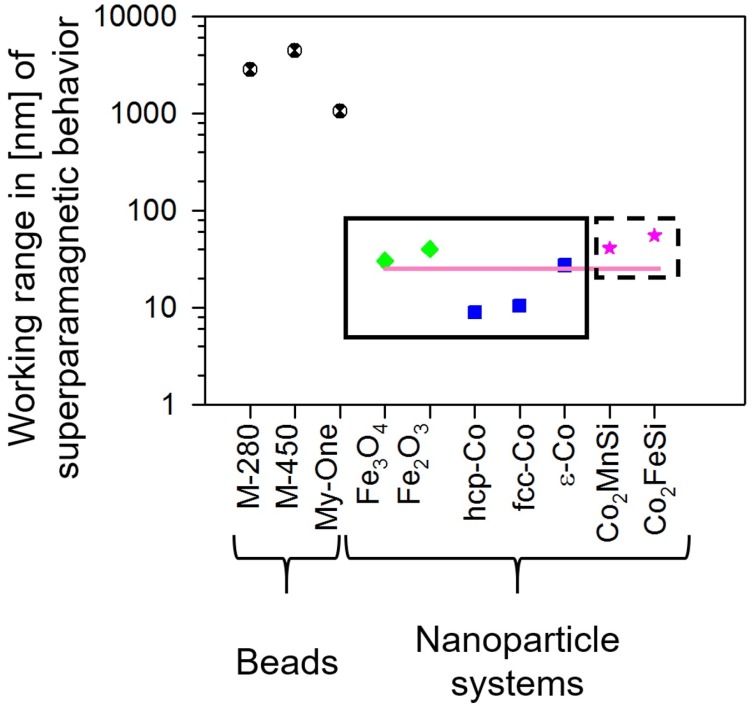
Comparison of the working range of possible superparamagnetic phases to serve as nanoparticles for labelling and separation of polysomes.

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
