# Peer review of "Magnetic Tracking of Protein Synthesis in Microfluidic Environments—Challenges and Perspectives"

_nanomaterials, 2019, doi:10.3390/nano9040585_

Round 1
Reviewer 1 Report
General Comments: The authors present a compelling idea to isolate polysome components from lysed cells via magnetic nanoparticles in microfluidic devices to track protein synthesis. While the general concept appears to be novel (and heroic if accomplished), I have several concerns regarding its efficacy. Given that this is a perspective and is thus intended to spur discussions for future work, I believe this article merits publication after the following concerns have been addressed.
Major Concern: My main concern stems from a lack of understanding, perhaps due to my background in colloid and interface science (not molecular biology). Do the authors plan to track polypeptide growth after isolation of polysome components? If so, a discussion of this is completely absent from the text. Alternatively, do the authors plan to isolate polysomes from cell lysate and quantify? If so, then I believe the title of the article is misleading. In my mind, it suggests a method for tracking peptide synthesis in real time. Such a “static” approach will only provide a snapshot of various translational structures, which can give some statistical insights, but feedback on translational kinetics. Therefore, I ask that the authors clarify their approach in the text so that readers will better understand and appreciate what is proposed.
If the method is indeed dynamic, the authors should discuss how tagging various polysome species with small magnetic nanoparticles would slow the rate of translation (e.g., by slowing the diffusivity, and thus rate, of polysome subunit assembly). Further, depending on where magnetic species are bound, the assembly and function of polysomes will likely be compromised (e.g., via disruption of the initiation phase). A discussion of this and other possibilities is critical. Further, the sizes of the antibodies used to bind nanoparticles are often as large as the subunit components themselves. How do the authors propose to do this without compromising translation?
However, if this method is static, I ask that the authors expand the Challenges and Perspectives Section to weigh the advantages and disadvantages of this method compared to BONCAT, FUNCAT, SUnSET, TRICK, QuaNCAT, etc.—some of which seem to enable real time observation. How is this approach better? What are its limits? Second, I ask that the authors discuss how quantification will be performed (and what information this will provide). Third, I suggest that the authors revise the title to avoid misleading readers.
Minor Comments:
Regarding Figure 2B: to which ribosomal subunit does anti-S14 bind? Also, please increase the text size above the scale bar.
How will the authors determine which nanoparticles bind to which polysome targets (e.g., large ribosome vs. mRNA)? Is this important for analysis? Please discuss.
Microfluidic devices are popular, in part, due to their controlled laminar flow. In this regime, convective flow dominates over diffusive transport. Yet, binding magnetic beads to polysome subunits requires mixing, which presumably relies on their diffusion. This is a challenge for microfluidics and is solved by slowing flow rates substantially or incorporating active mixing elements. Please discuss how this will be accomplished and what throughputs are expected (i.e., flow rates or single cell lysate processed per time).
If a third type of bead is to bind to the nascent polypeptide (as shown in Figure 2C), where in the microfluidic device schematic would this be added?
Due to the extreme sensitivity of magnetic ratchets, how do the authors propose preventing fouling (i.e., deposition of waste, debris, particles) over time?
How will the ratchet be fabricated? Please add brief details and references to established methods for readers who may be unfamiliar.
Author Response
Major Concern: My main concern stems from a lack of understanding, perhaps due to my background in colloid and interface science (not molecular biology). Do the authors plan to track polypeptide growth after isolation of polysome components? If so, a discussion of this is completely absent from the text.
We apologize for this ambiguity and have clarified the issue further. As pointed out before, the polysome is arrested during extraction by addition of the inhibitor cycloheximide. This is needed for stabilization. Nevertheless, the order of the ribosome within the polysome also carries dynamic information since with progression along the transcript, the nascent protein expands. We propose to label nascent proteins with magnetic beads in order to separate polysomes using magnetic gradients or magnetic ratchets.
Alternatively, do the authors plan to isolate polysomes from cell lysate and quantify? If so, then I believe the title of the article is misleading. In my mind, it suggests a method for tracking peptide synthesis in real time. Such a “static” approach will only provide a snapshot of various translational structures, which can give some statistical insights, but feedback on translational kinetics. Therefore, I ask that the authors clarify their approach in the text so that readers will better understand and appreciate what is proposed.
We use “protein synthesis” as synonym for the translational process in general not specifically for the growth of the peptide chain. Therefore, we would like to keep the title with some addition as suggested by reviewer 2. To prevent misunderstandings we highlight in our article that the method will not allow for dynamic measurements but to determine the arrested states of translation (l. 282 and following).
If the method is indeed dynamic, the authors should discuss how tagging various polysome species with small magnetic nanoparticles would slow the rate of translation (e.g., by slowing the diffusivity, and thus rate, of polysome subunit assembly). Further, depending on where magnetic species are bound, the assembly and function of polysomes will likely be compromised (e.g., via disruption of the initiation phase). A discussion of this and other possibilities is critical.
As commented above the proposed method is not meant to be dynamic. We suggest to use cycloheximide as an efficient translational inhibitor to stall ribosomes on the mRNA during cell lysis and thus to ‘freeze’ the current translational state. All ongoing dynamic translational processes will be arrested. We added this part in l. 204-208.
Further, the sizes of the antibodies used to bind nanoparticles are often as large as the subunit components themselves. How do the authors propose to do this without compromising translation?
Thanks for the concern of the referee which would only be valid if we aimed to study the translational dynamics. But as outlined before, cycloheximide will stall translation prior to analysis. Single domain antibodies, so called nanobodies, or aptamers are discussed as alternative probes to address nascent proteins (l. 178-181).
However, if this method is static, I ask that the authors expand the Challenges and Perspectives Section to weigh the advantages and disadvantages of this method compared to BONCAT, FUNCAT, SUnSET, TRICK, QuaNCAT, etc.—some of which seem to enable real time observation. How is this approach better? What are its limits?
We expanded the “Challenges and Perspectives” section in order to provide a detailed discussion of advantages of the proposed methods and its novelty compared with other previously described and applied approaches.
Second, I ask that the authors discuss how quantification will be performed (and what information this will provide).
The magnetic ratchet separates the labelled polysomes along the microfluidic channel depending on the numbers of ribosomes. Pre trials with standardized molecules will give hints after what time molecules of which size reach which positions. Outlets in the magnetic ratchet will enable the harvesting of polysomes. Afterwards, the molecules can be quantified by biochemical/ molecular biological approaches like qPCR, absorption or by physical sensors. A detailed explanation was added in the manuscript in l. 232 and following.
Third, I suggest that the authors revise the title to avoid misleading readers.
We included a detailed discussion about what this method could provide or not. This should clarify all potential misunderstandings.
Minor Comments:
Regarding Figure 2B: to which ribosomal subunit does anti-S14 bind? Also, please increase the text size above the scale bar.
We added to the text that anti S14 binds the ribosomal protein S14 of the small ribosomal subunit. The text size above the scale bar was increased.
How will the authors determine which nanoparticles bind to which polysome targets (e.g., large ribosome vs. mRNA)? Is this important for analysis? Please discuss.
As suggested before in our manuscript the different labelling strategies will be applied in a consecutive manner. For example: After cell lysis and cycloheximide treatment DNA-functionalized nanoparticles will be introduced into the system in order to label specific transcripts by complementary hybridization. A magnetic gradient is then used to separate the magnetic beads, decorated with polysomes from undecorated ones, unlabelled “waste” and cell debris. The next steps would be the IgG labelling of ribosomes and the magnetic ratchet where polysomes carrying different numbers of ribosomes will be separated from each other. Detectors will count the delivered assemblies for each channel. The output would be how many transcripts of a certain gene carry one ribosome, how many carry 5 ribosomes, how many 10 and so on. We added a detailed explanation for this in l. 237- 239 and emphasized the optional combination of different magnetic beads once more.
Microfluidic devices are popular, in part, due to their controlled laminar flow. In this regime, convective flow dominates over diffusive transport. Yet, binding magnetic beads to polysome subunits requires mixing, which presumably relies on their diffusion. This is a challenge for microfluidics and is solved by slowing flow rates substantially or incorporating active mixing elements. Please discuss how this will be accomplished and what throughputs are expected (i.e., flow rates or single cell lysate processed per time).
We have addressed the issue of mixing in detail. Moreover, our proposed microfluidic device contains reaction chambers to enhance the bonding between magnetic nanoparticles and polysomes.
If a third type of bead is to bind to the nascent polypeptide (as shown in Figure 2C), where in the microfluidic device schematic would this be added?
When describing the different labelling strategies we suggest to combine two of them and apply them successively. The type of probes depends on the aim of study. Optionally one can add a third type of probe. We included an “optional inlet” in Figure 3 and described that the labelling strategies can be combined as required and do not have to be applied together.
Due to the extreme sensitivity of magnetic ratchets, how do the authors propose preventing fouling (i.e., deposition of waste, debris, particles) over time?
Due to the stepwise labelling unlabelled molecules and cell debris will be separated from magnetic beads by magnetic gradients as indicated in Figure 3. We emphasized this once more in l. 235-237.
How will the ratchet be fabricated? Please add brief details and references to established methods for readers who may be unfamiliar.
The fabrication process with all successive steps has now been integrated within our manuscript.
Reviewer 2 Report
Magnetic Tracking of Protein Synthesis (Nanomaterials, MDPI)
This is a Perspective manuscript that proposes a completely new approach to study protein synthesis, by using functionalized superparamagnetic particles in a microfluidic environment. The manuscript is very well written, especially, the sections devoted to protein synthesis and the existing methods for analyzing the synthesis of proteins in vitro and in cells. These parts are well documented, the existing methods are described and the need for novel tools is emphasized.
Taking into account the content of this paper, it is suggested to change the title to:
Magnetic Tracking of Protein Synthesis in a Microfluidic Environment. Challenges and Perspectives, which would better reflect the “perspective” nature of the work.
It is recommended to expand the advantages of the microfluidic approach, not in general, but for this particular work. An additional figure, illustrating potential superparamagnetic nanoparticles having attached polysomes, in a microfluidic device, would be helpful to show how the separation would work. .
Author Response
Reviewer 2:
This is a Perspective manuscript that proposes a completely new approach to study protein synthesis, by using functionalized superparamagnetic particles in a microfluidic environment. The manuscript is very well written, especially, the sections devoted to protein synthesis and the existing methods for analyzing the synthesis of proteins in vitro and in cells. These parts are well documented, the existing methods are described and the need for novel tools is emphasized.
Taking into account the content of this paper, it is suggested to change the title to:
Magnetic Tracking of Protein Synthesis in a Microfluidic Environment. Challenges and Perspectives, which would better reflect the “perspective” nature of the work.
The title of the review was adjusted to” Magnetic Tracking of Protein Synthesis in Microfluidic Environments – Challenges and Perspectives”
It is recommended to expand the advantages of the microfluidic approach, not in general, but for this particular work.
Advantages of microfluidic devices for this application are highlighted in the Challenges and Perspective-part.
An additional figure, illustrating potential superparamagnetic nanoparticles having attached polysomes, in a microfluidic device, would be helpful to show how the separation would work.
Since we are discussing a first approach to reach our goal of magnetic tracking of protein synthesis we feel that the degree of details at this point is sufficient and in agreement with our experimental experience. During the forthcoming implementation of our device and resulting first measurements we are more than willing to share further details with the community.
Reviewer 3 Report
The authors outline a concept for combining magnetic beads with microfluidics to separate protein synthesis complexes using different binding affinity strategies. This is a clever idea that would be able to help shed new light on the protein synthesis process if reduced to practice. This would be great to demonstrate in practice. As a perspective paper, this is a good concept and should be pursued. However, there are several parts that are highly speculative and need additional literature citations in order to demonstrate the feasibility of the overall approach as described below. In Figure 2C, how can antibodies be created to specifically bind to the growing protein? Usually the selection process for antibodies uses fully formed proteins that are static and have a specific and stable 3D conformation. Since the proteins are being produced in a dynamic process how can antibodies be made to be specific for the protein in this state? In figure 6 the authors show that the ideal nanoparticles would be less than 40nm in diameter. Will magnet beads in this size range be able to generate enough force in a magnetic field to cause successful separation of these large protein complexes in a flowing microfluidic environment? The authors need to cite literature that shows that the forces generated on magnetic particles in the desired size range are large enough to cause the type of separation they suggest would be necessary for the microfluidic separation. The required magnetic fields need to be of reasonable magnitude and spatial resolution for this application. Additional Comments: It would be helpful for the authors to suggest more concrete microfluidic channel designs that would enable the type of serration they envision in addition to the overall concept diagram shown in figure 3. How exactly are the beads and the cell lysates going to be mixed to ensure good binding opportunities? Usually straight channels will cause laminar flow and slow down any mixing. The authors show some basic concept for the magnetic ratchet concept in Figure 3 and 4 but it would be good to see how this would all be combined in a real chip design. Will the lysis buffers interfere with the different affinity binding strategies downstream or will they need to wash away or otherwise remove the lysis buffer components in the channel flow? What dimensions of the channels will be needed for single cell applications to make sure the lysates are not diluted beyond a useful range through the microfluidic flow process? Will these dimensions be feasible to fabricate? In figure 3, will the addition of magnetic particles to both the ribosomes and the RNA itself cause preferential separation due to the magnetic-force/complex-mass ratio not changing as additional subunits are added? If you only have 1 magnetic particle per complex but complexes of varying mass then you would be able to separate based on mass.Author Response
The authors outline a concept for combining magnetic beads with microfluidics to separate protein synthesis complexes using different binding affinity strategies. This is a clever idea that would be able to help shed new light on the protein synthesis process if reduced to practice. This would be great to demonstrate in practice. As a perspective paper, this is a good concept and should be pursued. However, there are several parts that are highly speculative and need additional literature citations in order to demonstrate the feasibility of the overall approach as described below. In Figure 2C, how can antibodies be created to specifically bind to the growing protein? Usually the selection process for antibodies uses fully formed proteins that are static and have a specific and stable 3D conformation. Since the proteins are being produced in a dynamic process how can antibodies be made to be specific for the protein in this state?
We have expanded this section to deal with this point. It is described starting in l. 275: “Polyclonal antibodies prepared against denatured proteins should contain IgGs that recognize epitopes of the nascent protein. Alternatively, epitope-specific antibodies should be generated against peptide stretches of the N-terminus.” We also point out the potential of nanobodies and aptamers.
In figure 6 the authors show that the ideal nanoparticles would be less than 40nm in diameter. Will magnet beads in this size range be able to generate enough force in a magnetic field to cause successful separation of these large protein complexes in a flowing microfluidic environment? The authors need to cite literature that shows that the forces generated on magnetic particles in the desired size range are large enough to cause the type of separation they suggest would be necessary for the microfluidic separation. The required magnetic fields need to be of reasonable magnitude and spatial resolution for this application.
We have added a section on an alternative route to fabricate nanoparticles, which are superparamagnetic and have a magnetization, which is about twice of that of magnetite.
Additional Comments: It would be helpful for the authors to suggest more concrete microfluidic channel designs that would enable the type of serration they envision in addition to the overall concept diagram shown in figure 3.
The channel design has now been concretized.
How exactly are the beads and the cell lysates going to be mixed to ensure good binding opportunities? Usually straight channels will cause laminar flow and slow down any mixing. The authors show some basic concept for the magnetic ratchet concept in Figure 3 and 4 but it would be good to see how this would all be combined in a real chip design.
The mixing is now addressed in detail.
Will the lysis buffers interfere with the different affinity binding strategies downstream or will they need to wash away or otherwise remove the lysis buffer components in the channel flow?
Due to our experience we do not expect any negative influence of buffer components which the bonding between nanoparticles and polysomes.
What dimensions of the channels will be needed for single cell applications to make sure the lysates are not diluted beyond a useful range through the microfluidic flow process? Will these dimensions be feasible to fabricate?
We have introduced the channel dimension and discussed the reasoning behind. From our experience these dimensions will be feasible so as to realize the polysomes tracking and separation.
In figure 3, will the addition of magnetic particles to both the ribosomes and the RNA itself cause preferential separation due to the magnetic-force/complex-mass ratio not changing as additional subunits are added? If you only have 1 magnetic particle per complex but complexes of varying mass then you would be able to separate based on mass.
We have added a short discussion of this point starting in l. 296: “The experimental design will have to be optimized, e.g., one can chose between labelling with several superparamagnetic nanoparticles or a single nanoparticle combined with unmagnetic labels. The selection of the labelling strategy will have to be harmonized with the separation and detection method used in the microfluidic design.”
Round 2
Reviewer 1 Report
I thank the authors for their careful response to my concerns and lapses in understanding. I am satisfied with the current version of the manuscript, and I recommend it for publication barring the following changes:
Can the authors clarify in the abstract that ribosomal translation is arrested during the extraction and quantification steps? Also, can the authors indicate in the abstract that their approach to gain insights into the dynamics of translation is through quantifying large numbers of ribosomes along different locations of the polysome? While it may seem obvious to the authors, this will help non-experts greatly.
Author Response
We have added the two requested clarifications in the manuscript. They are highlighted in the uploaded version of the manuscript. Thanks to this and all other referees for their constructive and knowledgable comments.
Round 3
Reviewer 1 Report
The authors have addressed all concerns and suggestions.